# Determinants of transthyretin levels and their association with adverse clinical outcomes among UK Biobank participants

Naman S. Shetty[1,2] ✉, Mokshad Gaonkar[3,4], Nirav Patel[3], Akhil Pampana ®[3], Nehal Vekariya[3], Peng Li ®[5], Garima Arora[3] & Pankaj Arora ®[3,6] ✉

Transthyretin is a transport protein whose misfolding has been implicated in the development of cardiac amyloidosis. Here, we examine the clinical correlates of transthyretin levels, the differences in transthyretin levels according to the pathogenic V142I *TTR* variant carrier status, and the association of transthyretin levels with outcomes among 35,206 UK Biobank participants who underwent plasma profiling and were free from prevalent cardiovascular disease and chronic renal disease. Transthyretin levels are lower in females, decrease with increasing C-reactive protein levels, and increase with body mass index, systolic blood pressure, diastolic blood pressure, total cholesterol, albumin levels, triglyceride levels, and creatinine levels. V142I non-carriers [n = 35,167, mean: −0.1 (0.3)] have higher adjusted transthyretin levels compared with the carriers [n = 39, mean: −0.5 (0.3)] (p:<0.001). A standard deviation decrease in transthyretin levels increases the risk of heart failure [HR_{adj}: 1.17 (95% Confidence Interval = 1.08–1.26)] and all-cause mortality [HR_{adj}: 1.18 (95% Confidence Interval = 1.14–1.24)]. This study shows that individuals with low transthyretin levels, such as those carrying the V142I variant, are at a higher risk of heart failure and mortality.

Transthyretin (TTR), a homotetrameric protein synthesized in the liver, is responsible for transporting thyroxine and retinol[1]. Dissociation and misfolding of the TTR protein leads to the formation of amyloid fibrils[1]. Transthyretin cardiac amyloidosis, secondary to the deposition of these amyloid fibrils, is characterized by reduced compliance of the heart and an increased risk of heart failure and death[1,2]. TTR misfolding may be accentuated in the presence of a TTR genetic mutation leading to greater TTR instability, which multiplies the risk of outcomes. Considering the centrality of TTR instability in the pathogenesis of cardiac amyloidosis, TTR levels have been used as a therapeutic target to evaluate the efficacy of disease-modifying therapy for TTR cardiac amyloidosis[3]. However, comprehensive data regarding the determinants of TTR levels, the impact of specific genetic variants such

as the V142I *TTR* gene variant on TTR levels, and the association of TTR levels with clinical outcomes are lacking. This study leveraged data from the UK Biobank to examine the clinical correlates of TTR levels, the differences in TTR levels based on the V142I *TTR* gene variant carrier status, and the association of TTR levels with outcomes.

In this work, we demonstrate that TTR levels are lower in females compared with males, increase with higher body mass index (BMI), systolic blood pressure (SBP), diastolic blood pressure (DBP), total cholesterol, albumin levels, triglyceride levels, and creatinine levels, and reduce with increasing C-reactive protein (CRP) levels. The study finds that individuals carrying the V142I *TTR* gene variant have lower TTR levels compared with non-carriers. Furthermore, lower TTR levels are associated with an increased risk of heart failure,

[1]Department of Anesthesia, Critical Care and Pain Medicine, Massachusetts General Hospital, Boston, MA, USA. [2]Harvard Medical School, Boston, MA, USA. [3]Division of Cardiovascular Disease, University of Alabama at Birmingham, Birmingham, AL, USA. [4]Department of Biostatistics, University of Alabama at Birmingham, Birmingham, AL, USA. [5]School of Nursing, University of Alabama at Birmingham, Birmingham, AL, USA. [6]Section of Cardiology, Birmingham Veterans Affairs Medical Center, Birmingham, AL, USA. ✉e-mail: nshetty3@mgb.org; parora@uabmc.edu

**Table 1 | Baseline characteristics stratified by V142I TTR variant carrier status**

|  | Overall (n = 35,206) | V142I carrier (n = 39) | V142I non-carrier (n = 35,167) |
|---|---|---|---|
| Age, y | 58.0 (50.0, 63.0) | 51.0 (45.0, 58.0) | 58.0 (45.0, 58.0) |
| Females | 19,399 (55.1) | 23 (59.0) | 19,376 (55.1) |
| Systolic blood pressure, mmHg | 137.7 (18.5) | 137.2 (19.0) | 137.7 (18.5) |
| Diastolic blood pressure, mmHg | 82.2 (10.2) | 84.7 (10.5) | 82.2 (10.2) |
| Body mass index, kg/m² | 27.3 (4.7) | 30.0 (5.9) | 27.3 (4.7) |
| Townsend deprivation Index[a] | −2.1 (−3.6, 0.7) | 3.3 (−0.2, 6.8) | −2.1 (−3.6, 0.7) |
| Transthyretin levels, NPX[b] | −0.1 (0.3) | −0.5 (0.3) | −0.1 (0.3) |

[a]Townsend deprivation index is a measure of socioeconomic deprivation that takes car ownership, house ownership, household overcrowding, and employment into consideration. A higher value indicates higher socioeconomic deprivation.
[b]*NPX* normalized protein expression. NPX refers to the unit of quantification for analytes measured on the Olink platform. The NPX values are measured on the $\log_2$ scale, with lower values indicating lower levels of the analyte.

cardiovascular disease, atherosclerotic cardiovascular disease, all-cause mortality, and cardiovascular mortality. The study results emphasize the need to define therapeutic thresholds for TTR levels while considering the factors known to affect TTR levels.

## Results

### Baseline characteristics
Among 35,206 individuals with complete data included in the study, the median age was 58 (50, 63) years and 19,399 (55.1%) were females. (Table 1) TTR levels were lower in females [β = −0.100 (95% Confidence Interval = −0.107 to −0.09)], decreased with CRP levels [β = −0.060 (95% Confidence Interval = −0.063 to −0.057) per 1.1 mg/dL] and increased with BMI [β = 0.010 (95% Confidence Interval = 0.007 to 0.013) per 4.7 kg/m²], SBP [β = 0.012 (95% Confidence Interval = 0.008 to 0.019) per 18.5 mmHg], DBP [β = 0.008 (95% Confidence Interval = 0.005 to 0.012) per 10.2 mmHg], total cholesterol [β = 0.032 (95% Confidence Interval = 0.029 to 0.034) per 43.8 mg/dL], albumin levels [β = 0.061 (95% Confidence Interval = 0.059 to 0.064) per 0.1 mg/dL], triglyceride levels [β = 0.066 (95% Confidence Interval = 0.063 to 0.069) per 0.5 mg/dL], and creatinine levels [β = 0.027 (95% Confidence Interval = 0.024 to 0.030) per 0.2 mg/dL]. (Table 2) The adjusted mean TTR levels were 9.6% (95% Confidence Interval = 9.0–10.1%) lower in females compared with males. (p <0.001) (Fig. 1) V142I carriers [n = 39, mean: −0.5 (0.3)] had lower adjusted TTR levels compared with the non-carriers [n = 35,167, mean: −0.1 (0.3)] (p < 0.001). (Fig. 2)

### Association of transthyretin levels with heart failure
There were 31,429 individuals with follow-up data in the study. Among the 31,429 individuals, the median age was 58 (50, 64) years and 17,421 (55.4%) were females. Over the median follow-up of 13.7 (13.0–14.4) years, there were 401 (2.6%) and 494 (3.1%) events of heart failure in individuals with high and low TTR levels, respectively. The incidence rate of heart failure was 1.92 (95% Confidence Interval = 1.74–2.12) per 1,000 person-years and 2.39 (95% Confidence Interval = 2.19–2.62) per 1000 person-years in individuals with high and low TTR levels, respectively. (Table 3) The risk of heart failure increased by 17% per SD decrease in TTR levels [HR$_{adj}$: 1.17 (95% Confidence Interval = 1.08–1.26)]. (Figs. 3, 4).

### Association of transthyretin levels with all-cause mortality
During the study period, there were 1347 (8.6%) and 1774 (11.3%) events of all-cause mortality in individuals with high and low TTR levels, respectively. The incidence rate of all-cause mortality was 6.40 (95% CI: 6.07–6.75) per 1000 person-years in individuals with high TTR levels and 8.53 (95% CI: 8.14–8.93) per 1000 person-years in individuals with low TTR levels. (Table 3) The risk of all-cause mortality increased by

**Table 2 | Association of clinical correlates with transthyretin levels**

| Correlates of TTR | Standardized regression coefficients | P value |
|---|---|---|
| Female sex | −0.100 (−0.107 to −0.09) | <0.001 |
| Age (per 8.2 years) | 0.000 (−0.002 to 0.003) | 0.25 |
| Body mass index (per 4.7 kg/m²) | 0.010 (0.007 to 0.013) | <0.001 |
| Diastolic blood pressure (per 10.2 mmHg) | 0.008 (0.005 to 0.012) | <0.001 |
| Systolic blood pressure (per 18.5 mmHg) | 0.012 (0.008 to 0.016) | <0.001 |
| Total cholesterol (per 43.8 mg/dL) | 0.032 (0.029 to 0.034) | <0.001 |
| Log albumin (per 0.1 mg/dL) | 0.061 (0.059 to 0.064) | <0.001 |
| Log triglyceride (per 0.5 mg/dL) | 0.066 (0.063 to 0.069) | <0.001 |
| Log creatinine (per 0.2 mg/dL) | 0.027 (0.024 to 0.030) | <0.001 |
| Log hsCRP (per 1.1 mg/dL) | −0.060 (−0.063 to −0.057) | <0.001 |

The p-value depicted was obtained from linear regression models.

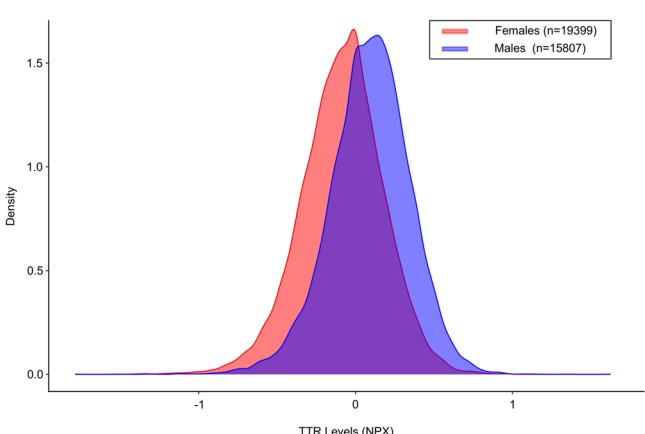

**Fig. 1 | Transthyretin levels by sex.** This figure demonstrates the transthyretin (TTR) levels stratified by sex. Females (n = 19,399) and males (n = 15,807) are presented in red and blue, respectively.

18% per SD decrease in TTR levels. [HR$_{adj}$: 1.18 (95% Confidence Interval = 1.14–1.24)] (Figs. 4, 5)

### Association of transthyretin levels with cardiovascular mortality
For the outcome of cardiovascular mortality, there were 159 (1.0%) events in individuals with high TTR levels and 250 (1.6%) in individuals

with low TTR levels over the follow-up period. The incidence rate of cardiovascular mortality was 0.76 (95% CI: 0.64–0.89) per 1000 person-years and 1.20 (95% CI: 1.05–1.37) per 1000 person-years in individuals with high and low TTR levels, respectively. (Table 3) The adjusted hazard ratio for cardiovascular mortality was 1.33 (95% Confidence Interval = 1.19–1.49) per SD decrease in TTR levels. (Figs. 4, 5).

### Association of transthyretin levels with atherosclerotic cardiovascular disease

For the outcome of atherosclerotic cardiovascular disease, there were 1,187 (7.6%) events in individuals with high TTR levels and 1291 (8.2%) events in individuals with low TTR levels during the follow-up period. The incidence rate of atherosclerotic cardiovascular disease was 5.82 (95% CI: 5.50–6.16) per 1000 person-years and 6.42 (95% CI: 6.08–6.78) per 1000 person-years in individuals with high and low TTR levels, respectively. (Table 3) The adjusted hazard ratio of atherosclerotic

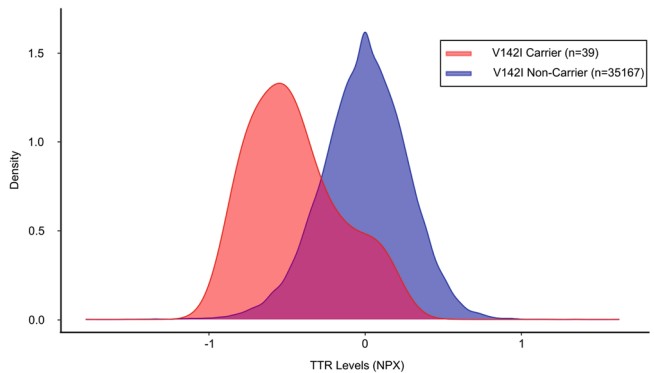

**Fig. 2 | Transthyretin levels by V142I carrier status.** This figure demonstrates the transthyretin (TTR) levels stratified by the carrier status for the V142I *TTR* gene variant. V142I carriers (n = 39) and non-carriers (n = 35,167) are presented in red and blue, respectively.

cardiovascular disease was 1.14 (95% Confidence Interval = 1.09–1.20) per SD decrease in TTR levels. (Figs. 4, 5)

### Association of transthyretin levels with cardiovascular disease

Over the follow-up period, individuals with high and low TTR levels had 1,458 (9.3%) and 1,622 (10.3%) events of cardiovascular disease, respectively. The incidence rate of cardiovascular disease was 7.19 (95% CI: 6.83–7.57) per 1000 person-years in individuals with high TTR levels and 8.12 (95% CI: 7.73–8.53) per 1000 person-years in individuals with low TTR levels. (Table 3) The risk of cardiovascular disease increased by 14% per SD decrease in TTR levels. [HR$_{adj}$: 1.14 (95% Confidence Interval = 1.10–1.19)] (Figs. 4, 5)

## Discussion

This large study, including ~35,000 individuals with TTR levels, demonstrated that TTR levels were significantly lower in females compared with males. TTR levels were noted to be negatively associated with CRP levels and positively associated with BMI, SBP, DBP, total cholesterol, albumin levels, triglyceride levels, and creatinine levels. Asymptomatic carriers of the V142I *TTR* gene variant were observed to have lower levels of TTR compared with V142I non-carriers. Longitudinal analysis showed that individuals with low TTR levels were at a higher risk of heart failure and other adverse cardiovascular outcomes. To summarize, this study highlights that individuals with low TTR levels, such as those carrying the V142I *TTR* variant, are at an increased risk of cardiovascular outcomes and mortality.

Examination of the correlates of TTR levels demonstrated two notable correlates that reduced TTR levels i.e., female sex and CRP levels. While the precise mechanism of sex differences in TTR levels has not been ascertained, it could be postulated that sex hormones may play a role in altering TTR production. Sex hormones have been shown to regulate hepatic TTR expression in animal models, with testosterone inducing a larger increase in TTR synthesis than estrogen[4]. Notably, CRP was shown to be negatively associated with TTR levels. This finding could indicate that an increase in oxidative

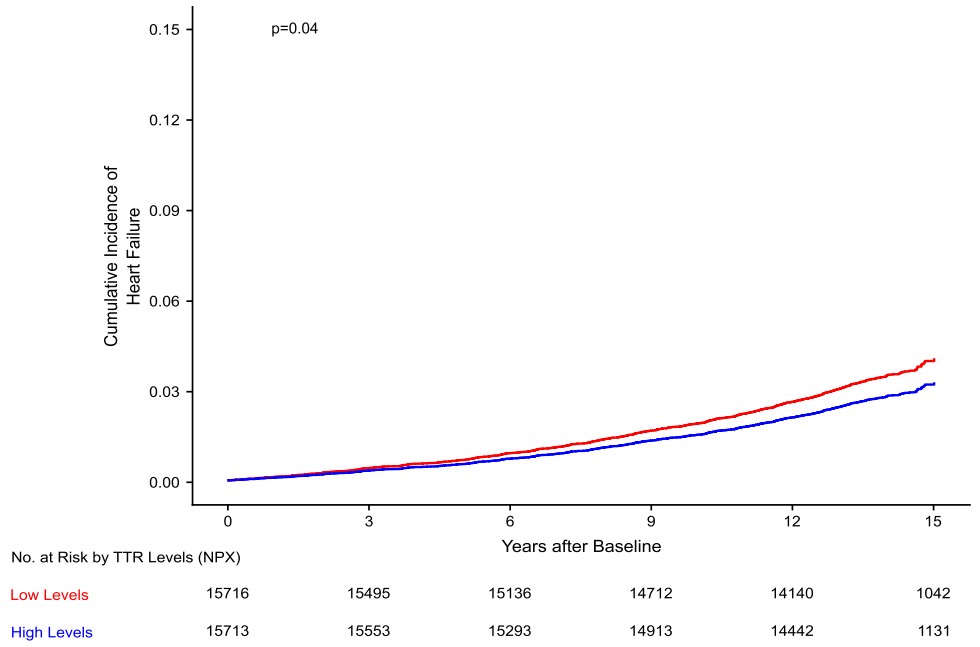

**Fig. 3 | Cumulative incidence of heart failure stratified by transthyretin levels.** This figure depicts the cumulative incidence curves for heart failure stratified by transthyretin (TTR) levels. The cumulative incidence curves were generated using the Kaplan–Meier method. Based on TTR levels, sex-specific high (≥median) and low (<median) TTR groups were created. Low and high TTR groups are depicted in red and blue, respectively. The p-value comparing the TTR groups was derived using the log rank test.

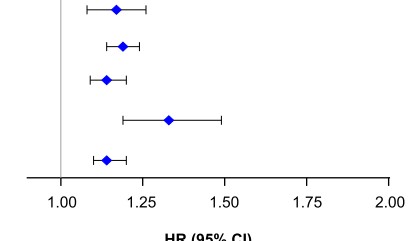

| Outcome | HR (95% CI) |
|---|---|
| Heart Failure | 1.17 (1.08-1.26) |
| All-Cause Mortality | 1.18 (1.14-1.24) |
| Cardiovascular Mortality | 1.33 (1.19-1.49) |
| Atherosclerotic Cardiovascular Disease | 1.14 (1.09-1.20) |
| Cardiovascular Disease | 1.14 (1.10-1.19) |

**Fig. 4 | Association of transthyretin levels with outcomes.** This figure depicts a forest plot summarizing the association of transthyretin (TTR) levels and the study outcomes [heart failure, cardiovascular disease (composite of heart failure, stroke, and coronary heart disease), atherosclerotic cardiovascular disease (composite of stroke and coronary heart disease), all-cause mortality, and cardiovascular mortality among 35,206 individuals. Cox models, adjusted for age, sex, Townsend index, BMI, SBP, total cholesterol, diabetes, smoking status, eGFR, statin use, and antihypertensive use, were used to examine the association of TTR levels and the study outcomes. The hazard ratio and 95% confidence interval per SD decrease in TTR levels have been depicted.

stress and inflammation may promote the destabilization of the TTR protein[5].

Prior literature examining the correlates and prognostic value of TTR levels focused on individuals who had already developed wild-type cardiac amyloidosis[6,7]. Individuals with wild-type cardiac amyloidosis were found to have similar TTR levels as controls without cardiac amyloidosis[6]. Among individuals with wild-type cardiac amyloidosis, low TTR levels were found to predict the risk of overall survival over a median follow-up of ~3 years[7]. Given that these previous studies were limited in sample size and conducted in individuals with disease[6,7], the current study adds to the literature by examining the determinants of TTR levels in a large population of healthy individuals. The presence of low TTR levels in V142I carriers compared with non-carriers previously reported in a study with a small population was reaffirmed in the current study[6]. Furthermore, this study examined the prognostic value of TTR levels in healthy individuals over a long follow-up duration. Concordant with prior literature[7], low TTR levels were found to be associated with an increased risk of all-cause mortality. Apart from overall survival, this study demonstrated that low TTR levels also increase the risk of additional adverse clinical outcomes such as heart failure, cardiovascular mortality, cardiovascular disease, and atherosclerotic cardiovascular disease. Although the underlying mechanism by which reduced TTR levels increase the risk of adverse outcomes is not known, it could be postulated that decreased TTR levels reflect TTR protein instability. Prior literature has demonstrated that individuals carrying the V142I TTR destabilizing genetic variant have an increased risk of heart failure due to deposition of misfolded TTR fibrils in the heart[2,8–10]. Therefore, individuals with lower TTR levels could be hypothesized to have higher TTR instability, which in turn, increases their risk of adverse clinical outcomes.

This study has several public health implications. Carrying the V142I *TTR* variant has been associated with an increased risk of heart failure and mortality[2,8–12]. TTR protein stabilizing therapy such as Tafamidis in patients with transthyretin cardiac amyloidosis has been shown to increase TTR levels by stabilizing the native tetrameric structure of TTR and decreasing the risk of mortality[3,13]. However, the link between the V142I carrier status, TTR levels, and clinical outcomes had not been explored prior to the current study. This study supports that V142I carriers have lower TTR levels compared with non-carriers and low TTR levels are associated with an increased risk of heart failure. Incorporation of TTR levels in the screening programs for hereditary cardiac amyloidosis may be especially beneficial considering that the V142I variant is a common genetic variant with a carrier prevalence of 3–4% among Black individuals[14–16]. Considering the late age of phenotypic presentation in V142I carriers, serial TTR level measurement in family members who are asymptomatic V142I carriers may be a useful risk stratification tool to estimate their risk of developing heart failure and guide preventive measures.

This study had several limitations. First, the possibility of a selection bias exists due to the exclusion of individuals with missing data. Second, given the observation nature of this study, the associations reported in the study may be affected by residual confounding and unmeasured confounders. Third, considering the e-value for the observed associations in the current study, minimal confounding could potentially nullify these relationships. Fourth, the UK Biobank recruited predominantly White individuals above 40 years of age, which precludes the generalizability of the study findings to other populations. Fifth, the current study could not account for the diurnal changes in TTR levels[17]. Lastly, the use of ICD codes to define outcomes may be prone to misclassification.

In conclusion, the current study elucidates the factors influencing TTR levels and demonstrates that individuals with low TTR have a higher risk of adverse clinical outcomes. This study draws attention to

### Table 3 | Association of study outcomes with transthyretin levels

| | Low transthyretin levels (n = 15,716) | High transthyretin levels (n = 15,713) | p-value |
|---|---|---|---|
| **Heart failure** | | | |
| Events | 494 (3.1) | 401 (2.6) | |
| Incidence rate | 2.39 (95% CI: 2.19–2.62) | 1.92 (95% CI: 1.74–2.12) | 0.001 |
| **All-Cause mortality** | | | |
| Events | 1774 (11.3) | 1347 (8.6) | |
| Incidence rate | 8.53 (95% CI: 8.14–8.93) | 6.40 (95% CI: 6.07–6.75) | <0.001 |
| **Cardiovascular mortality** | | | |
| Events | 250 (1.6) | 159 (1.0) | |
| Incidence rate | 1.20 (95% CI: 1.05–1.37) | 0.76 (95% CI: 0.64–0.89) | <0.001 |
| **Atherosclerotic cardiovascular disease** | | | |
| Events | 1291 (8.2) | 1187 (7.6) | |
| Incidence rate | 6.42 (95% CI: 6.08–6.78) | 5.82 (95% CI: 5.50–6.16) | 0.01 |
| **Cardiovascular disease** | | | |
| Events | 1622 (10.3) | 1458 (9.3) | |
| Incidence rate | 8.12 (95% CI: 7.73–8.53) | 7.19 (95% CI: 6.83–7.57) | <0.001 |

The p-value for the difference in two incidence rates was obtained using the Chi-squared statistic.

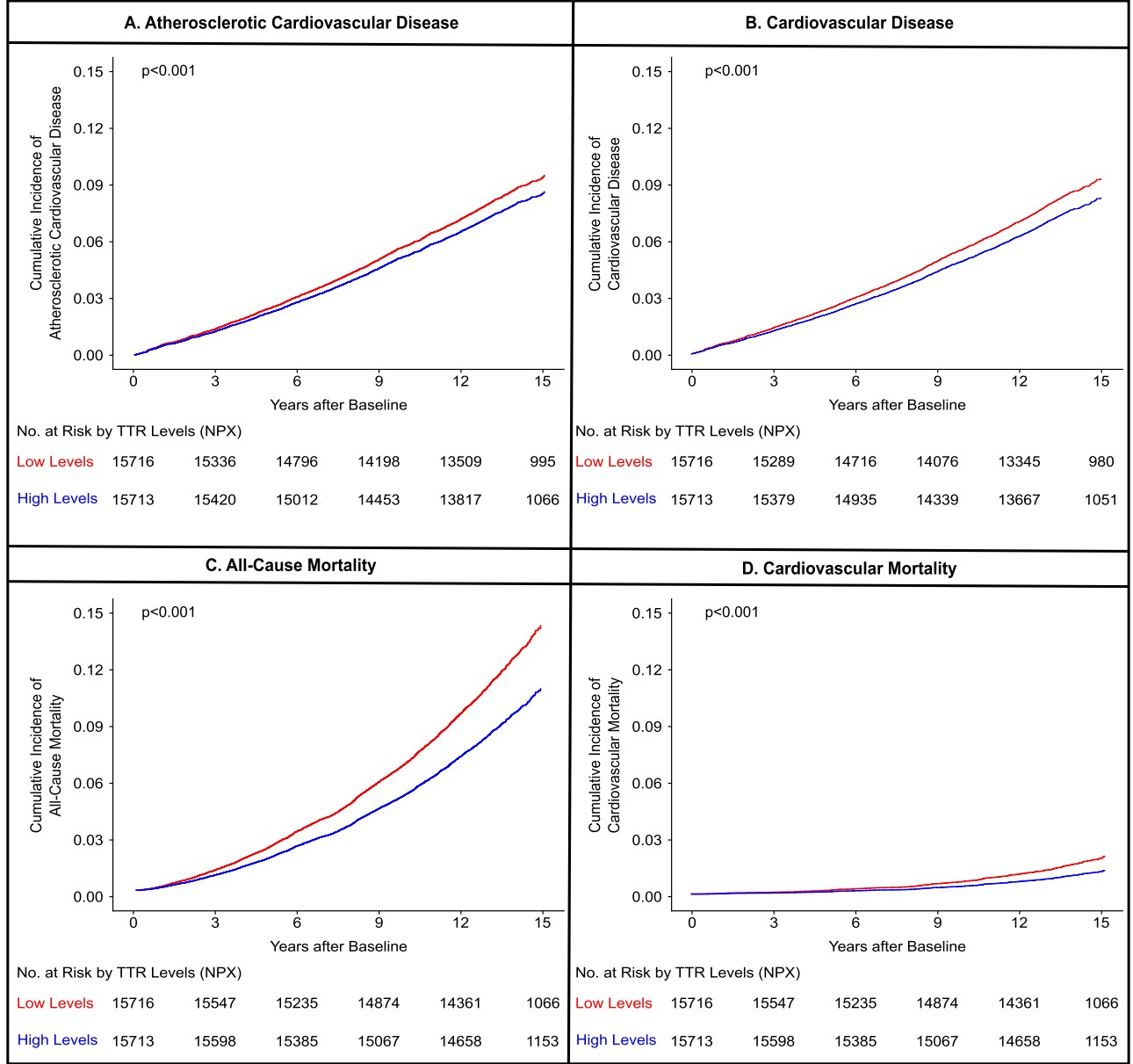

**Fig. 5 | Cumulative incidence of secondary outcomes stratified by transthyretin levels.** This figure depicts the cumulative incidence curves for atherosclerotic cardiovascular disease (composite of stroke and coronary heart disease) (**A**), cardiovascular disease (composite of heart failure, stroke, and coronary heart disease) (**B**), all-cause mortality (**C**), and cardiovascular mortality (**D**) stratified by TTR levels. The cumulative incidence curves were generated using the Kaplan-Meier method. Based on TTR levels, sex-specific high (≥median) and low (<median) TTR groups were created. Low and high TTR groups are depicted in red and blue, respectively. The p-value comparing the TTR groups was derived using the log rank test.

the need to recognize the factors affecting TTR levels and establish precise therapeutic thresholds for TTR levels, taking these correlates into consideration. Future studies are required to investigate the mechanisms underlying the association of the risk of adverse clinical outcomes in individuals with low TTR levels.

## Methods
### Study population
Data from the UK Biobank was used for the current study[18]. The UK Biobank was approved by the North West Centre for Research Ethics Committee (REC No. 16/NW/0274). The UK Biobank is a prospective cohort initiated in 2006 to examine the role of genetic and non-genetic factors in adults aged between 40 and 69 years. A total of 502,493 participants were recruited from 22 centers across the UK. Each

participant completed a series of surveys to gather information on lifestyle and health. They also underwent anthropometry, physical examination, and biological sample collection. Additionally, data from the electronic health records were collected from consenting participants. Written informed consent was provided by each participant.

Among the 502,493 participants from the UK Biobank, 54,219 participants from the UK Biobank underwent plasma profiling under the UK Biobank Plasma Profiling Project (UKBB-PPP). Participants were selected through a stratified approach that took into account age, sex, and recruitment center, as previously detailed, for plasma profiling[19]. The baseline characteristics of the randomly selected participants were representative of the overall UK Biobank cohort. This study included participants from the UK Biobank who underwent plasma profiling. Individuals with prevalent heart failure, coronary heart

disease (CHD), stroke, chronic kidney disease, and pregnant females were excluded from this analysis.

## Measurement of transthyretin levels

TTR levels were measured on the Olink Explore 3072, as described previously[19]. Briefly, the UKBB-PPP utilized plasma samples that were separated from blood specimens collected in EDTA tubes and stored at −80 °C[19]. Plasma samples were thawed and plated using the TECAN freedomEVO liquid handler before being shipped[19]. These plasma samples were shipped on dry ice to the Olink Analysis Service in Sweden for analysis[19]. Samples were analyzed using the Olink Explore 3072, which is an antibody-based proximity extension assay[19]. The Olink Explore 3072 platform was used to measure 2923 unique proteins across 8 protein panels[19]. Extensive quality control steps were implemented[19]. TTR levels measured on the Olink platform were converted to a $\log_2$ scale and quantified using Normalized Protein eXpression (NPX) values.

## Genotyping

This study utilized whole exome sequencing (WES) data from the UK Biobank was used to identify carriers of the *TTR* V142I pathogenic variant. Briefly, WES was conducted on the Illumina NovaSeq 6000 sequencer with a coverage of 20× or greater at the Regeneron Genetics Center[20,21]. The exomes were aligned to the hg38 reference genome using BWA-MEM. Sample level and variant level quality control steps were implemented.

## Outcomes

The primary outcome was heart failure and secondary outcomes included all-cause mortality, cardiovascular mortality, atherosclerotic cardiovascular disease (composite of CHD and stroke), and cardiovascular disease (composite of heart failure, stroke, and CHD). Heart failure was selected as the primary outcome for this study, taking into account the TTR tetramer destabilization in carriers of the pathogenic V142I variant and the association of the pathogenic V142I variant and heart failure[2,8–10]. Outcomes were identified using self-reported data, Hospital Episode Statistics, and death registries. Validated International Classification of Diseases Ninth and Tenth Revision Codes were used to identify outcomes for this study[22,23]. (Supplementary Table 1) The time to event for each outcome was determined by calculating the duration between the date of the event and the date on which blood samples were collected for plasma profiling. For individuals who did not have an event, the time to event was censored at death or the last date on which data was available.

## Statistical analysis

Linear regression models regressing TTR levels on age, sex, SBP, DBP, BMI, CRP levels, triglyceride levels, albumin levels, creatinine levels, and total cholesterol levels were used to examine the association of TTR levels with the clinical correlates. Given the skewed distribution of CRP levels, triglyceride levels, albumin levels, and creatinine levels, these variables were log-transformed. These covariates were selected based on prior literature suggesting an association of the variable with TTR levels[24,25]. Continuous variables in the linear regression model were standardized to allow comparison across variables.

TTR levels by sex and V142I *TTR* variant carrier status were compared with adjustment for age, sex, BMI, SBP, DBP, CRP levels, triglyceride levels, albumin levels, creatinine levels, and total cholesterol levels. Considering the sex differences in the TTR levels observed in the study, sex-specific median values were used to categorize the cohort into individuals with high TTR levels (≥median) and low TTR levels (<median). Poisson regression was used to determine the incidence rate for each outcome in individuals with high and low TTR levels. Kaplan Meier curves were generated for each study outcome in individuals with high and low TTR levels. The log-rank test was used to

examine the difference in the cumulative incidence of outcomes by TTR levels. Multivariable-adjusted Cox models were used to estimate the adjusted hazard ratio of study outcomes per SD decrease in TTR levels. All analyses were conducted on SAS 9.4 (SAS Institute, Cary, NC).

## Reporting summary

Further information on research design is available in the Nature Portfolio Reporting Summary linked to this article.

## Data availability

The data supporting the findings from this study are available within the manuscript. The research has been conducted using the UK Biobank Resource under Application Number 58838. Eligible researchers may access UK Biobank data by application on www.ukbiobank.ac.uk. Source data are provided with this paper.

## Code availability

Code for data analysis performed in the study can found on the following URL: https://github.com/Arora-Translational-Lab/TTR_Nature_Comm_2024_shetty_et_al/blob/main/Code.

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

## Acknowledgements
Pankaj Arora is supported by the National Heart, Lung, and Blood. Institute of the National Institutes of Health (NIH) awards (R01HL160982, R01HL163852, and R01HL163081). Nirav Patel is supported by the National Institutes of Health grant T32HL007457.

## Author contributions
P. A. conceptualized and designed the study. P. A., N. S. S., M. G., A. P., N. P., N. V., P. L., and G. A. acquired, analyzed, or interpreted data. P. A., N. S. S., M. G., A. P., N. P., N. V., P. L., and G. A. drafted the manuscript. All authors performed critical revisions of the manuscript. M. G. and A. P. did the statistical analysis. P. A. and P. L. supervised and verified the data analysis.

## Competing interests
Pankaj Arora reports grant support from Merck Sharp & Dohme LLC and Bristol-Myers Squibb and consulting income from Bristol-Myers Squibb, which are all unrelated to this work. The other authors declare no competing interests.
