## [Peer Review File · Nature Communications]

REVIEWER COMMENTS

Reviewer #1 (Remarks to the Author):

Title: Transthyretin Levels, Clinical Correlates, and Adverse Clinical Outcomes

Summary: Using the UK Biobank, the authors investigated the association between TTR levels and an incident diagnosis of heart failure, and a secondary outcome of all-cause mortality. The authors conclude “that individuals with low TTR levels, such as those carrying the V142I variant, are at a higher risk of heart failure and mortality. The authors conclude “that individuals with low TTR levels, such as those carrying the V142I variant, are at a higher risk of heart failure and mortality.”

General comments: I was requested to “focus on those aspects related to the applied statistical methods.”

Before commenting on the statistical methods, one needs to understand the study and target populations. The authors are making inferences about the US population, ($n > 300\text{MM}$) based on the UK Biobank population (UK Biobank, $n > 0.5\text{MM}$) from which a sample ($n=5421$) was identified which was reduced to a complete case sample ($n=35206$) and a final sample ($n=31429$) with follow-up data. These various reductions in the sample population do not necessarily arise randomly. In other words, the possibility of selection bias can't be discounted. Missing data is another concern and complete case analyses are known to be a suboptimal manner to deal with missing variables.

The authors state their first statistical analysis is a linear regression of TTR levels with 5 clinical variables (age, sex, systolic blood pressure (SBP), diastolic blood pressure (DBP), and body mass index (BMI)). Their choice of the multivariable linear model is not justified and the assumptions underlying the use of linear regression don't appear to have been tested. How does one know that the relationships identified would not change or disappear if other potential confounders were introduced into the model?

The Poisson and Cox models apparently don't use continuous TTR values but rather a dichotomized version which results in a loss of information. Again justification for model selection, model verification and diagnostics seem to be missing. The authors also neglect to discuss that besides selection bias and missing data a small amount of confounding ($evalue = 1.3$) could negate their findings.

Reviewer #2 (Remarks to the Author):

The present manuscript by Dr. Naman S. Shetty and colleagues offers intriguing new insights into the relationship between TTR levels and long-term outcomes based on data from the UK Biobank registry. TTR levels were evaluated in patients initially free from cardiovascular (coronary heart disease, heart failure, stroke), renal disease, and pregnancy. Median TTR levels were analyzed by genotype (V142I and non-V142I carriers) and gender, with prognosis ultimately described by high or low levels of TTR concentration.

The sample size, comprising over 30,000 patients, is deemed appropriate, and the results are hypothesis-generating, as different patient subgroups may exhibit varying TTR cutoff values, potentially impacting long-term outcomes. Please find major comments below:

- Introduction: No major comments.

- Methods section:

I. The choice of heart failure as the primary outcome, rather than overall survival or cardiovascular disease-free survival, should be further discussed.

II. A more detailed description of median TTR levels across gender and age brackets (especially below and above 50 or 60 years of age) would benefit readers. Are there differing cutoff values depending on when TTR is measured?

III. Follow-up determination should be better described.

IV. While the population was assessed for the specific vATTR mutation (p.V142I), consideration could be given to screening for other genotypes (T60A, V30M, etc.), which could be collectively grouped as non-p.V142I carriers.

V. Is it possible to determine the number of patients who developed amyloidosis based on disease codes? This could be particularly relevant for variant carriers, as pointed out by the authors in the Discussion section.

- Results section:

I. What was the median follow-up for the entire cohort? Additional analyses by gender, genotype, and age brackets would be clinically insightful.

II. What was the median time to event among patients? Are there differences across gender, genotype, and age at TTR determination?

III. Was Table 2 adjusted for genotype?

IV. Please report p-values for incidence rate comparisons between low and high levels of TTR (Table 3) and for Kaplan-Meier curves.

- Discussion:

I. While this study utilizes data from the UK Biobank, it would be valuable to compare the results with previous evidence in ATTR amyloidosis (e.g., <https://doi.org/10.1161/CIRCHEARTFAILURE.117.004000>, [10.1080/13506120802525285](https://doi.org/10.1080/13506120802525285)) to identify additional strengths of the present work.

- Abstract:

Define acronyms.

Reviewer #3 (Remarks to the Author):

Response to Review

Reviewer #1 (Remarks to the Author):

Title: Transthyretin Levels, Clinical Correlates, and Adverse Clinical Outcomes

Summary: Using the UK Biobank, the authors investigated the association between TTR levels and an incident diagnosis of heart failure, and a secondary outcome of all-cause mortality. The authors conclude “that individuals with low TTR levels, such as those carrying the V142I variant, are at a higher risk of heart failure and mortality. The authors conclude “that individuals with low TTR levels, such as those carrying the V142I variant, are at a higher risk of heart failure and mortality.”

General comments: I was requested to “focus on those aspects related to the applied statistical methods.”

Before commenting on the statistical methods, one needs to understand the study and target populations. The authors are making inferences about the US population, (n>300MM) based on the UK Biobank population (UK Biobank, n > 0.5MM) from which a sample (n=5421) was identified which was reduced to a complete case sample (n=35206) and a final sample (n=31429) with follow-up data. These various reductions in the sample population do not necessarily arise randomly. In other words, the possibility of selection bias can't be discounted.

Response: We appreciate the Reviewer's comment. The inferences to the US population in the discussion were made based on the prevalence of the pathogenic TTR gene variant (V142I) that was examined in the current study. Several studies, including our prior work, have shown that the population prevalence of the V142I variant among Black individuals in the US is ~3.4%.¹⁻⁵ However, as the Reviewer suggests, the results from the current study in a UK-based population should be generalized to the US population with caution. Therefore, we have removed the sentence in the discussion section on extrapolating our findings to the US population.

As the Reviewer notes, the UK Biobank participants were excluded in a step-wise manner in our study. The exclusion of individuals in our study was done to add rigor to our analysis. Our study aimed to characterize the correlates of transthyretin levels in a healthy population and to test the hypothesis that low transthyretin levels predicted the risk of heart failure and other adverse cardiovascular events. Therefore, we excluded any factors that could affect transthyretin levels, such as individuals with prevalent heart failure, coronary heart disease, stroke, chronic kidney disease, and pregnant females. The exclusion of individuals with the relevant factors is standard practice in studies examining the clinical correlates of an analyte.⁶⁻¹⁰

The current study was conducted in the Plasma Proteomics Project subsample of the UK Biobank.¹¹ The majority of the participants (n=46,595) were randomly selected from the baseline visit. A small proportion of participants were selected by the UKB-PPP consortium members (n=6,376), and 1,268 participants were selected from the COVID-19 repeat imaging

study. The randomly selected participants were demographically representative of the overall UK Biobank cohort except for a higher Townsend index. **Reviewer Figure 1** depicts the derivation of the study population.

Reviewer Figure 1: Study Population Flowchart

Similarly, we compared the baseline characteristics of all individuals with TTR levels measured, the cohort in which the clinical correlates of TTR levels were examined, and the cohort for the longitudinal analysis. We found that the baseline characteristics were similar in our study cohorts and the cohort of individuals with TTR measurements. (**Reviewer Table 1**)

Reviewer Table 1: Baseline Characteristics in Individuals with Transthyretin Level Measurements, Individuals with Transthyretin Level Measurements After Applying Exclusion Criteria, and Individuals with Transthyretin Level Measurements After Applying Exclusion Criteria with Longitudinal Data

	Cohort with TTR Measurements (n=44,369)	Cohort with TTR Measurements Applying Exclusion Criteria (n=35,206)	Cohort with TTR Measurements Applying Exclusion Criteria with Longitudinal Data (n=31,429)
Age, y	58.0 (50.0-64.0)	58.0 (50.0-63.0)	58.0 (50.0-64.0)
Females	23,986 (54.1%)	19,399 (55.1%)	17,405 (55.4%)
Systolic Blood Pressure, mmHg	137.8 (18.6)	137.7 (18.5)	137.9 (18.5)
Diastolic Blood Pressure, mmHg	82.0 (10.3)	82.2 (10.2)	82.2 (10.2)
Body mass index, kg/m²	27.5 (4.8)	27.3 (4.7)	27.5 (4.7)
Townsend Deprivation Index*	-2.0 (-3.6-0.8)	-2.1 (-3.6-0.7)	-2.1 (-3.6-0.7)
Transthyretin Levels, NPX**	-0.1 (0.3)	-0.1 (0.3)	-0.1 (0.3)

However, as noted by the Reviewer, we recognize that a selection bias may occur due to the exclusion of individuals from our study. We have highlighted this limitation of our study in the discussion section of our revised manuscript.

Page 4, Line 84

Among the 502,493 participants from the UK Biobank, 54,219 participants from the UK Biobank underwent plasma profiling under the UK Biobank Plasma Profiling Project (UKBB-PPP). Participants were selected through a stratified approach that took into account age, sex, and recruitment center, as previously detailed, for plasma profiling.¹¹

Page 11, Line 231

First, the possibility of a selection bias exists due to the exclusion of individuals with missing data.

Missing data is another concern and complete case analyses are known to be a suboptimal manner to deal with missing variables.

Response: We appreciate the Reviewer’s comment. A complete case analysis was conducted due to two reasons: 1) to avoid making untestable assumptions about missingness mechanisms, and 2) the relatively small proportion of missingness in the longitudinal cohort will not affect the

efficiency of the analysis. However, we do agree with the Reviewer that a complete case analysis may not be an optimal method to deal with missing covariates in some cases. Therefore, we repeated all analyses in our study using multiple imputations, assuming missingness at random.

Data for missing covariates in the longitudinal analysis was imputed. The fully conditional specification method for multiple imputation was used.¹² We created 10 separate imputed datasets and then analyzed each dataset independently. Estimates from independent datasets were pooled using Rubin’s rules using PROC MIANALYZE. The multiple imputation model included sex, race, age, Townsend deprivation index, body mass index, systolic blood pressure, estimated glomerular filtration rate, lipid-lowering medication use, antihypertensive medication use, total cholesterol levels, smoking status, and diabetes status. Data for diabetes, race, smoking status, total cholesterol levels, and systolic blood pressure was imputed for 1,527, 152, 167, 2,619, and 961 individuals, respectively. The association of TTR levels with the study outcomes has been presented in **Reviewer Table 2**. The complete case analysis and multiple imputation-based analysis demonstrated similar effect estimates for the association of TTR levels with study outcomes.

Reviewer Table 2: Comparison of the Risk of Study Outcomes Per Standard Deviation Decrease in Transthyretin Levels in the Complete Case Analysis and Multiple Imputation-Based Analysis

Outcome	Complete Case Analysis	Multiple Imputation
Heart Failure	1.17 (1.08-1.26)	1.16 (1.08-1.25)
Cardiovascular Disease	1.14 (1.10-1.19)	1.13 (1.08-1.17)
Atherosclerotic Cardiovascular Disease	1.14 (1.09-1.20)	1.12 (1.07-1.17)
All-Cause Mortality	1.18 (1.14-1.24)	1.20 (1.16-1.25)
Cardiovascular Mortality	1.33 (1.19-1.49)	1.36 (1.22-1.51)

The authors state their first statistical analysis is a linear regression of TTR levels with 5 clinical variables (age, sex, systolic blood pressure (SBP), diastolic blood pressure (DBP), and body mass index (BMI)). Their choice of the multivariable linear model is not justified and the assumptions underlying the use of linear regression don’t appear to have been tested. How does one know that the relationships identified would not change or disappear if other potential confounders were introduced into the model?

Response: We thank the Reviewer for their comment. The selection of covariates was based on clinically relevant variables that are likely to influence TTR levels. An extensive literature review revealed additional factors that may play a role in determining TTR levels. Therefore, we have included these additional covariates in our linear regression models. The additional

covariates included in our revision are CRP levels, triglyceride levels, albumin levels, creatinine levels, and total cholesterol levels.¹³⁻¹⁵ Given the skewed distribution of CRP levels, triglyceride levels, albumin levels, and creatinine levels, these variables were log-transformed for further analyses. The inclusion of these additional covariates is likely to reduce potential confounding.

First, we examined the association of each covariate with TTR levels using univariate models. Variables that were statistically significant were carried forward in the multivariable model. In the univariate and multivariable models, we found that sex, body mass index, systolic blood pressure, diastolic blood pressure, CRP levels, triglyceride levels, albumin levels, creatinine levels, and total cholesterol levels remained statistically significant. (**Reviewer Table 3**)

Reviewer Table 3: Univariate and Multivariate Association of Clinical Correlates with Transthyretin Levels

Correlates of TTR	Univariate Analysis		Multivariate Analysis	
	Standardized Regression Coefficients	P value	Standardized Regression Coefficients	P value
Female Sex	-0.177 (-0.183 to -0.172)	<0.001	-0.100 (-0.107 to -0.09)	<0.001
Age (per 8.2 years)	0.002 (-0.001 to 0.005)	0.14	0.000 (-0.002 to 0.003)	0.25
Body Mass Index (per 4.7 kg/m ²)	0.004 (0.001 to 0.007)	0.001	0.010 (0.007 to 0.013)	<0.001
Diastolic Blood Pressure (per 10.2 mmHg)	0.043 (0.040 to 0.046)	<0.001	0.008 (0.005 to 0.012)	<0.001
Systolic Blood Pressure (per 18.5 mmHg)	0.041 (0.039 to 0.044)	<0.001	0.012 (0.008 to 0.016)	<0.001
Total Cholesterol (per 43.8 mg/dL)	0.047 (0.044 to 0.049)	<0.001	0.032 (0.029 to 0.034)	<0.001
Log Albumin (per 0.1 mg/dL)	0.090 (0.087 to 0.093)	<0.001	0.061 (0.059 to 0.064)	<0.001
Log Triglyceride (per 0.5 mg/dL)	0.084 (0.081 to 0.087)	<0.001	0.066 (0.063 to 0.069)	<0.001
Log Creatinine (per 0.2 mg/dL)	0.065 (0.063 to 0.068)	<0.001	0.027 (0.024 to 0.030)	<0.001
Log hsCRP (per 1.1 mg/dL)	-0.051 (-0.053 to -0.048)	<0.001	-0.060 (-0.063 to -0.057)	<0.001

However, we acknowledge that the observational nature of our study prevents us from accounting for all potential confounders, particularly for those unmeasured or unknown confounders. We have highlighted these limitations in the limitation section of our manuscript.

Page 6, Line 121

Linear regression models regressing TTR levels on age, sex, systolic blood pressure (SBP), diastolic blood pressure (DBP), body mass index (BMI), CRP levels, triglyceride levels, albumin levels, creatinine levels, and total cholesterol levels were used to examine the association of TTR levels with the clinical correlates. Given the skewed distribution of CRP levels, triglyceride levels, albumin levels, and creatinine levels, these variables were log-transformed. These covariates were selected based on prior literature suggesting an association of the variable with TTR levels.^{13,14}

Page 11, Line 232

Second, given the observation nature of this study, the associations reported in the study may be affected by residual confounding and unmeasured confounders.

The Poisson and Cox models apparently don't use continuous TTR values but rather a dichotomized version which results in a loss of information. Again justification for model selection, model verification and diagnostics seem to be missing. The authors also neglect to discuss that besides selection bias and missing data a small amount of confounding (evalue =1.3) could negate their findings.

Response: We appreciate the Reviewer's comment. As per the Reviewer's suggestion, we have conducted our analyses using continuous TTR levels instead of the dichotomized variable. On using continuous TTR levels, we found that the adjusted hazard ratio of heart failure was 1.17 (95%CI: 1.08-1.26) per standard deviation (SD) decrease in TTR levels. Similarly, a per SD decrease in TTR levels increased the risk of all-cause mortality [HR_{adj}: 1.18 (95%CI: 1.14-1.24)], cardiovascular mortality [HR_{adj}: 1.33 (95%CI: 1.19-1.49)], atherosclerotic cardiovascular disease [HR_{adj}: 1.14 (95%CI: 1.09-1.20)], and cardiovascular disease [HR_{adj}: 1.14 (95%CI: 1.10-1.19)].

TTR levels were only dichotomized to depict the cumulative incidence of the study outcomes based on TTR levels. In our revised manuscript, we have clarified that the Cox models used continuous TTR levels.

As requested by the Reviewer, we have provided details on the model diagnostics below. In our linear regression models examining the correlates of TTR levels, we generated a plot of the residuals and the predicted TTR levels. (**Reviewer Figure 2**) The figure shows that there are no systemic patterns or deviations in the scatters, indicating a linear relationship. The random scatter of the residuals around the y-axis reference line indicates equal variance i.e. homoscedasticity across all levels of predictions.

Reviewer Figure 2: Residual Diagnostic Plot for Linear Regression Model

Reviewer Figure 3 depicts that the residuals from the linear regression model were normally distributed.

Reviewer Figure 3: Distribution of Residuals from Linear Regression Model

Lastly, we tested the presence of multicollinearity for the variables included in the linear regression models. The variance inflation factor was used to quantify multicollinearity. A variance inflation factor value above the threshold indicates the presence of multicollinearity.

Even when using a conservative threshold of 2.5, none of the variables included in the linear regression models demonstrated multicollinearity. (**Reviewer Table 4**)

Reviewer Table 4: Assessment of Multicollinearity Using the Variance Inflation Factor for the Linear Regression Models

Variable	Variance Inflation Factor
Sex	1.66
Age	1.26
Body Mass Index	1.42
Diastolic Blood Pressure	2.04
Systolic Blood Pressure	2.19
Total Cholesterol	1.20
Log Albumin	1.15
Log Triglyceride	1.33
Log Creatinine	1.52
Log hs-CRP	1.32

For our Cox models, we assessed the violation of the proportionality assumption. **Reviewer Figure 4** demonstrates the plot of Schoenfeld's residuals for the Cox model examining the association of heart failure with standardized TTR levels. The figure shows that the proportionality assumption for the Cox model was satisfied.

**Reviewer Figure 4: Schoenfeld's Residuals for the Cox Model
Examining the Association of Heart Failure with
Standardized Transthyretin Levels**

Furthermore, as suggested by the Reviewer, we have discussed the limitations of our study based on the e-value.

Page 6, Line 133

Poisson regression was used to determine the incidence rate for each outcome in individuals with high and low TTR levels. Kaplan Meier curves were generated for each study outcome in individuals with high and low TTR levels. Multivariable-adjusted Cox models were used to estimate the adjusted hazard ratio of study outcomes per SD decrease in TTR levels.

Page 11, Line 234

Third, considering the e-value for the observed associations in the current study, minimal confounding could potentially nullify these relationships.

Reviewer #2 (Remarks to the Author):

The present manuscript by Dr. Naman S. Shetty and colleagues offers intriguing new insights into the relationship between TTR levels and long-term outcomes based on data from the UK Biobank registry. TTR levels were evaluated in patients initially free from cardiovascular (coronary heart disease, heart failure, stroke), renal disease, and pregnancy. Median TTR levels were analyzed by genotype (V142I and non-V142I carriers) and gender, with prognosis ultimately described by high or low levels of TTR concentration.

The sample size, comprising over 30,000 patients, is deemed appropriate, and the results are hypothesis-generating, as different patient subgroups may exhibit varying TTR cutoff values, potentially impacting long-term outcomes. Please find major comments below:

- Introduction: No major comments.

- Methods section:

I. The choice of heart failure as the primary outcome, rather than overall survival or cardiovascular disease-free survival, should be further discussed.

Response: We thank the Reviewer for their guidance. The selection of heart failure as the primary outcome was motivated by the role of the TTR protein in the development of heart failure secondary to cardiac amyloidosis. We and others have shown that V142I variant carriers have an increased risk of heart failure.^{1-5,7} While the deposition of misfolded TTR levels in the heart has been implicated in the pathogenesis of heart failure, there is a lack of data on the effect of the V142I variant on TTR levels. Through our study, we aimed to establish the link between the V142I variant and the development of heart failure. An increase in mortality among variant carriers is primarily driven by an increase in cardiovascular mortality due to heart failure. Therefore, we selected heart failure as the primary outcome and mortality as the secondary outcome.

Page 5, Line 111

Heart failure was selected as the primary outcome for this study, taking into account the TTR tetramer destabilization in carriers of the pathogenic V142I variant and the well-established association of the pathogenic V142I variant and heart failure.²⁻⁵

II. A more detailed description of median TTR levels across gender and age brackets (especially below and above 50 or 60 years of age) would benefit readers. Are there differing cutoff values depending on when TTR is measured?

Response: We appreciate the Reviewer's comment. As recommended by the Reviewer, we analyzed the median TTR levels by age and sex. We found that the median TTR levels were higher in females. However, TTR levels did not vary by age group. (**Reviewer Table 5**)

Reviewer Table 5: Median Transthyretin Levels Stratified by Sex and Age

	Median Transthyretin Level
Sex	
Male	0.10 (-0.07, 0.26)
Female	-0.08 (-0.25, 0.08)
Age	
40-49 Years	-0.02 (-0.21, 0.17)
50-59 Years	0.02 (-0.16, 0.20)
60-69 Years	0.00 (-0.17, 0.17)

As the Reviewer astutely notes, TTR levels have been shown to have a diurnal rhythm.¹⁶ The current study could not account for the changes in TTR levels with time of day due to the lack of data. We have acknowledged this limitation in our revised manuscript.

Page 11, Line 237

Fifth, the current study could not account for the diurnal changes in TTR levels.¹⁶

III. Follow-up determination should be better described.

Response: We appreciate the Reviewer’s comment. In our UK Biobank-based analysis, outcome data was ascertained from multiple sources, including self-reported data, Hospital Episode Statistics, and death registries. Data from electronic health records was available till November 2022. The follow-up duration was determined based on the date of the event recorded in the aforementioned sources. The time-to-event was calculated by computing the days between the date of the event and the date of screening (during which blood samples were collected for estimating TTR levels). For individuals who did not have an event, their follow-up was censored on the last date for which electronic health record data was available.

Page 5, Line 116

The time to event for each outcome was determined by calculating the duration between the date of the event and the date on which blood samples were collected for plasma profiling. For individuals who did not have an event, the time to event was censored at death or the last date on which data was available.

IV. While the population was assessed for the specific vATTR mutation (p.V142I), consideration could be given to screening for other genotypes (T60A, V30M, etc.), which could be collectively grouped as non-p.V142I carriers.

Response: We thank the Reviewer for their suggestion. Based on the Reviewer’s suggestion, we analyzed the influence of additional TTR gene variants on circulating TTR levels. To generate a comprehensive list of TTR gene variants, we utilized TTR gene variants listed as pathogenic or likely pathogenic on ClinVar. The variants identified and their frequency has been summarized in **Reviewer Table 6**. Carriers of TTR variants other than the V142I variant were combined together to form a group of non-V142I TTR variant carriers.

Considering that the non-V142I are rare variants with a minor allele frequency <0.001, our study population had only 5 carriers of non-V142I variants. Therefore, further analyses were not conducted in non-V142I TTR variant carriers.

Reviewer Table 6: Distribution of Pathogenic/Likely Pathogenic TTR Gene Variants in the Overall UK Biobank Population and the Study Sample

Variants	Overall UKBB Population (n=454,756)	Study Sample (n=35,206)
Val142Ile	358 (0.1)	39 (0.1)
Val50Met	25 (0)	1 (0)
Thr60Ala	19 (0)	2 (0)
Thr79Ala	17 (0)	1 (0)
Ile88Leu	12 (0)	1 (0)
Ser70Gly	3 (0)	0 (0)
Ser97Tyr	3 (0)	0 (0)
Tyr136Ser	2 (0)	0 (0)
Ser43Asn	1 (0)	0 (0)
Glu81Lys	1 (0)	0 (0)
Tyr98Phe	1 (0)	0 (0)
Glu109Lys	1 (0)	0 (0)

V. Is it possible to determine the number of patients who developed amyloidosis based on disease codes? This could be particularly relevant for variant carriers, as pointed out by the authors in the Discussion section.

Response: We appreciate the Reviewer's comment. As per the Reviewer's suggestion, we examined the number of individuals who developed amyloidosis. Amyloidosis was identified based on validated International Classification of Diseases Ninth and Tenth Revision Codes (277.3 and E85).

To further examine if the incidence of amyloidosis varied by V142I carrier status, we leverage data from the overall UK Biobank cohort with data from 454,707 participants. In this large cohort, we found that 492 (0.1%) of 454,264 non-carriers and 16 (3.6%) of 443 V142I carriers developed amyloidosis. Therefore, V142I carriers were more likely to develop amyloidosis compared with non-carriers. In our cohort of 35,206 individuals, we found that 34 (0.1%) of 35,167 V142I non-carriers developed amyloidosis, while none of the 39 V142I carriers had the EHR disease code of amyloidosis.

However, we would like to highlight that amyloidosis is an underreported and underdiagnosed condition, which limits the utility of EHR-based data in conducting such analyses.

- Results section:

I. What was the median follow-up for the entire cohort? Additional analyses by gender, genotype, and age brackets would be clinically insightful.

Response: We thank the Reviewer for their suggestion. The median follow-up for our study sample was 13.7 (13.0-14.4) years.

As requested by the Reviewer, we conducted additional analyses to examine if the association of TTR levels with heart failure varied across the subgroup of gender, genotype, and age. We conducted interaction analyses to examine if there was a statistically significant interaction of TTR levels with gender, genotype, and age for the outcome of heart failure. We did not detect a statistically significant interaction of the TTR levels with the abovementioned factors for the outcome of heart failure. ($P_{\text{interaction}} > 0.10$) The risk of heart failure per SD decrease in TTR levels stratified by sex and age have been depicted in **Reviewer Table 7**. Analysis based on the V142I variant carrier was not conducted due to limited sample size.

Reviewer Table 7: Sex and Age stratified Hazard Ratio for Heart Failure

	Hazards Ratio (95% Confidence Interval)
Sex	
Male	1.22 (1.09-1.36)
Female	1.15 (1.01-1.32)
Age	
40-49 Years	1.16 (0.87-1.56)
50-59 Years	1.23 (1.03-1.48)
60-69 Years	1.20 (1.09-1.33)

II. What was the median time to event among patients? Are there differences across gender, genotype, and age at TTR determination?

Response: We appreciate the Reviewer’s thoughtful comment. The median time to event for heart failure was 9.2 (5.6-11.9) years.

As suggested by the Reviewer, we also examined if the time-to-event for heart failure varied across gender and age at TTR determination. We found that the time-to-event was similar in the subgroups of interest. (**Reviewer Table 8**) V142I carrier status-based median time to event has not been reported due to a lack of events in the group [most likely due to the small sample size (n=39) of the group].

Reviewer Table 8: Median Time to Event for Heart Failure Stratified by Sex and Age

	Median Time to Event (Interquartile Range)	P-Value
Sex		
Male	9.2 (5.4-12.0)	0.99
Female	9.2 (6.0-11.7)	
Age		
40-49 Years	10.6 (5.8-12.5)	0.08
50-59 Years	8.4 (5.2-11.4)	
60-69 Years	9.3 (5.9-12.0)	

III. Was Table 2 adjusted for genotype?

Response: We thank the Reviewer for their suggestion. Table 2 was not adjusted for genotype in our original analysis. In our revised manuscript, we have included additional clinical covariates to make our analysis more comprehensive, based on the feedback from **Reviewer 1**. These variables were selected based on their previously reported association with TTR levels. (**Reviewer Table 9**) As per the Reviewer's suggestion, we additionally adjusted for genotype in the analysis. Adjustment with genotype did not change the association of the clinical correlates with TTR levels.

Reviewer Table 9: Association of Clinical Correlates with Transthyretin Levels Adjusting for Genotype

Correlates of TTR	Standardized Regression Coefficients	P value
Female Sex	-0.100 (-0.107 to -0.09)	<0.001
Age (per 8.2 years)	0.000 (-0.002 to 0.003)	0.002
Body Mass Index (per 4.7 kg/m ²)	0.010 (0.007 to 0.013)	<0.001
Diastolic Blood Pressure (per 10.2 mmHg)	0.008 (0.005 to 0.012)	<0.001
Systolic Blood Pressure (per 18.5 mmHg)	0.012 (0.008 to 0.016)	<0.001
Total Cholesterol (per 43.8 mg/dL)	0.032 (0.029 to 0.034)	<0.001
Log Albumin (per 0.1 mg/dL)	0.061 (0.059 to 0.064)	<0.001
Log Triglyceride (per 0.5 mg/dL)	0.066 (0.063 to 0.069)	<0.001
Log Creatinine (per 0.2 mg/dL)	0.027 (0.024 to 0.030)	<0.001
Log hsCRP (per 1.1 mg/dL)	-0.060 (-0.063 to -0.057)	<0.001

IV. Please report p-values for incidence rate comparisons between low and high levels of TTR (Table 3) and for Kaplan-Meier curves.

Response: We are grateful for the Reviewer's comment. As suggested by the Reviewer, we have included the p-values for the comparison of the incidence rates of outcomes between individuals with low and high TTR levels. (**Reviewer Table 10**) We have also added the p-value to the Kaplan-Meier curves that were obtained from the log-rank test. (**Reviewer Figures 5 and 6**)

Page 6, Line 135

The log-rank test was used to examine the difference in the cumulative incidence of outcomes by TTR levels.

Reviewer Table 10: Incidence Rate of Study Outcomes Stratified by Transthyretin Levels

	Low Transthyretin Levels (n=15,716)	High Transthyretin Levels (n=15,713)	p-value
Heart Failure			
Events	494 (3.1)	401 (2.6)	
Incidence Rate	2.39 (95% CI: 2.19-2.62)	1.92 (95% CI: 1.74-2.12)	0.001
ASCVD			
Events	1,291 (8.2)	1,187 (7.6)	
Incidence Rate	6.42 (95% CI: 6.08-6.78)	5.82 (95% CI: 5.50-6.16)	0.01
CVD			
Events	1,622 (10.3)	1,458 (9.3)	
Incidence Rate	8.12 (95% CI: 7.73-8.53)	7.19 (95% CI: 6.83-7.57)	<0.001
All-Cause Mortality			
Events	1,774 (11.3)	1,347 (8.6)	
Incidence Rate	8.53 (95% CI: 8.14-8.93)	6.40 (95% CI: 6.07-6.75)	<0.001
Cardiovascular Mortality			
Events	250 (1.6)	159 (1.0)	
Incidence Rate	1.20 (95% CI: 1.05-1.37)	0.76 (95% CI: 0.64-0.89)	<0.001

Reviewer Figure 5: Cumulative Incidence of Heart Failure Stratified by Transthyretin Levels

No. at Risk by TTR Levels (NPX)

Low Levels	13953	13767	13845	13149	12687	979
High Levels	13971	13829	13582	13237	12801	934

Reviewer Figure 6: Cumulative Incidence of Study Outcomes Stratified by Transthyretin Levels

- Discussion:

I. While this study utilizes data from the UK Biobank, it would be valuable to compare the results with previous evidence in ATTR amyloidosis (e.g., <https://doi.org/10.1161/CIRCHEARTFAILURE.117.004000>, 10.1080/13506120802525285) to identify additional strengths of the present work.

Response: We thank the Reviewer for their suggestion. We read the manuscripts provided by the Reviewer with interest. Prior work examining TTR levels was predominantly focused on individuals with disease.

Buxbaum et al. examined TTR levels in 45 senile systemic amyloidosis patients with heart failure. Compared with age, sex, and race-matched controls, the TTR levels were similar in senile systemic amyloidosis patients with heart failure. In the control population, the TTR levels were found to be higher in males, Caucasians, and younger individuals compared with their respective counterparts. Additionally, the study demonstrated that carriers of the V142I TTR gene variant had lower TTR compared with V142V variant carriers in a cohort of African Americans.¹⁷

Similarly, Hanson et al. assessed the prognostic value of TTR levels among patients with biopsy-proven wild-type cardiac amyloidosis. In their cohort of 116 patients, they found that lower TTR levels were associated with decreased survival after accounting for left ventricular ejection fraction and elevated cTn-I levels. Furthermore, they demonstrated that TTR levels decreased in untreated patients with time over a 2-year follow-up period.¹⁸

While the previous studies focus on a diseased population, our study is among the first to examine the factors affecting TTR levels in a healthy population. Furthermore, unlike the previous studies that were conducted with a limited sample size, our study leverages data from ~35,000 individuals, making it the largest study to examine the determinants of TTR levels. Additionally, our study is enriched with a long follow-up duration and access to electronic health records allowed us to examine multiple outcomes, such as heart failure, cardiovascular disease, atherosclerotic cardiovascular disease, and cardiovascular mortality, in addition to overall survival.

Page 9, Line 202

Prior literature examining the correlates and prognostic value of TTR levels focused on individuals who had already developed wild-type cardiac amyloidosis.^{17,18} Individuals with wild-type cardiac amyloidosis were found to have similar TTR levels as controls without cardiac amyloidosis.¹⁷ Among individuals with wild-type cardiac amyloidosis, low TTR levels were found to predict the risk of overall survival over a median follow-up of ~3 years.¹⁸ Given that these previous studies were limited in sample size and conducted in individuals with disease^{17,18}, the current study adds to the literature by examining the determinants of TTR levels in a large population of healthy individuals. The presence of low TTR levels in V142I carriers compared with non-carriers that were previously reported in a small population was reaffirmed in the current study.¹⁷ Furthermore, this study examined the prognostic value of TTR levels in healthy

individuals over a long follow-up duration. Concordant with prior literature¹⁸, low TTR levels were found to be associated with an increased risk of all-cause mortality. Apart from overall survival, this study demonstrated that low TTR levels also increase the risk of additional adverse clinical outcomes such as heart failure, cardiovascular mortality, cardiovascular disease, and atherosclerotic cardiovascular disease.

- Abstract:

Define acronyms.

Response: We appreciate the Reviewer's comment. We have expanded the acronyms at the first instance of their use.

Abstract

Transthyretin (TTR) is a transport protein whose misfolding has been implicated in the development of cardiac amyloidosis. However, data on the clinical correlates of TTR levels, the differences in TTR levels according to the pathogenic V142I TTR gene variant carrier status, and the association of TTR levels with outcomes have not been assessed. Participants who underwent plasma profiling in the UK Biobank and who were free from prevalent cardiovascular disease and chronic renal disease were included in this study. TTR levels were measured on the Olink Explore 3072. The primary outcome was heart failure, and the secondary outcome was all-cause mortality. Multivariable adjusted regression models were used for analysis. Among the 35,206 individuals included, TTR levels were lower in females [β : -0.100 (95%CI: -0.107 to -0.09)], decreased with increasing CRP levels [β : -0.060 (95%CI: -0.063 to -0.057) per 1.1 mg/dL], and increased with systolic blood pressure [β : 0.012 (95%CI: 0.008 to 0.019) per 18.5 mmHg], diastolic blood pressure [β : 0.008 (95%CI: 0.005 to 0.012) per 10.2 mmHg], total cholesterol [β : 0.032 (95%CI: 0.029 to 0.034) per 43.8 mg/dL], albumin levels [β : 0.06 (95%CI: 0.059 to 0.064) per 0.1 mg/dL], triglyceride levels [β : 0.066 (95%CI: 0.063 to 0.069) per 0.5 mg/dL], and creatinine levels [β : 0.027 (95%CI: 0.024 to 0.030) per 0.2 mg/dL]. V142I non-carriers [n= 35,167, mean: -0.1 (0.3)] had higher adjusted TTR levels compared with the carriers [n= 39, mean: -0.5 (0.3)](p:<0.001). A standard deviation decrease in TTR levels was associated with an increase in the risk of heart failure [HR_{adj}: 1.17 (95%CI:1.08-1.26)] and all-cause mortality [HR_{adj}: 1.18 (95%CI: 1.14-1.24)]. This study shows that individuals with low TTR levels, such as those carrying the V142I variant, are at a higher risk of heart failure and mortality.

Reviewer #3 (Remarks to the Author):

Response: We thank the Reviewer for providing their valuable comments. We have incorporated the Reviewer's suggestion in our revised work.

References

- 1 Coniglio, A. C. *et al.* Transthyretin V142I Genetic Variant and Cardiac Remodeling, Injury, and Heart Failure Risk in Black Adults. *Jacc-Heart Fail* **10**, 129-138, doi:10.1016/j.jchf.2021.09.006 (2022).
- 2 Damrauer, S. M. *et al.* Association of the V122I Hereditary Transthyretin Amyloidosis Genetic Variant With Heart Failure Among Individuals of African or Hispanic/Latino Ancestry. *Jama-J Am Med Assoc* **322**, 2191-2202, doi:10.1001/jama.2019.17935 (2019).
- 3 Haring, B. *et al.* Cardiovascular Disease and Mortality in Black Women Carrying the Amyloidogenic V122I Transthyretin Gene Variant. *Jacc-Heart Fail* **11**, 1189-1199, doi:10.1016/j.jchf.2023.02.003 (2023).
- 4 Parcha, V. *et al.* Association of Transthyretin Val122Ile Variant With Incident Heart Failure Among Black Individuals. *Jama-J Am Med Assoc* **327**, 1368-1378, doi:10.1001/jama.2022.2896 (2022).
- 5 Quarta, C. C. *et al.* The Amyloidogenic V122I Transthyretin Variant in Elderly Black Americans. *New Engl J Med* **372**, 21-29, doi:10.1056/NEJMoa1404852 (2015).
- 6 Shetty, N. S. *et al.* Natriuretic Peptide Normative Levels and Deficiency The National Health and Nutrition Examination Survey. *Jacc-Heart Fail* **12**, 50-63, doi:10.1016/j.jchf.2023.07.018 (2024).
- 7 Fradley, M. G. *et al.* Reference Limits for N-Terminal-pro-B-Type Natriuretic Peptide in Healthy Individuals (from the Framingham Heart Study). *Am J Cardiol* **108**, 1341-1345, doi:10.1016/j.amjcard.2011.06.057 (2011).
- 8 Mu, S. *et al.* NT-proBNP Reference Intervals in Healthy U.S. Children, Adolescents, and Adults. *J Appl Lab Med* **8**, 700-712, doi:10.1093/jalm/jfad024 (2023).
- 9 McEvoy, J. W. *et al.* Myocardial Injury Thresholds for 4 High-Sensitivity Troponin Assays in a Population-Based Sample of US Children and Adolescents. *Circulation* **148**, 7-16, doi:10.1161/CIRCULATIONAHA.122.063281 (2023).
- 10 McEvoy, J. W. *et al.* Myocardial Injury Thresholds for 4 High-Sensitivity Troponin Assays in U.S. Adults. *J Am Coll Cardiol* **81**, 2028-2039, doi:10.1016/j.jacc.2023.03.403 (2023).
- 11 Sun, B. B. *et al.* Plasma proteomic associations with genetics and health in the UK Biobank. *Nature* **622**, 329-338, doi:10.1038/s41586-023-06592-6 (2023).
- 12 Van Buuren, S., Brand, J. P. L., Groothuis-Oudshoorn, C. G. M. & Rubin, D. B. Fully conditional specification in multivariate imputation. *J Stat Comput Sim* **76**, 1049-1064, doi:10.1080/10629360600810434 (2006).
- 13 Myron Johnson, A. *et al.* Clinical indications for plasma protein assays: transthyretin (prealbumin) in inflammation and malnutrition. *Clin Chem Lab Med* **45**, 419-426, doi:10.1515/CCLM.2007.051 (2007).
- 14 Kwanbunjan, K. *et al.* Association of retinol binding protein 4 and transthyretin with triglyceride levels and insulin resistance in rural thais with high type 2 diabetes risk. *BMC Endocr Disord* **18**, 26, doi:10.1186/s12902-018-0254-2 (2018).
- 15 Greve, A. M., Christoffersen, M., Frikke-Schmidt, R., Nordestgaard, B. G. & Tybjaerg-Hansen, A. Association of Low Plasma Transthyretin Concentration With Risk of Heart Failure in the General Population. *JAMA Cardiol* **6**, 258-266, doi:10.1001/jamacardio.2020.5969 (2021).

- 16 Fame, R. M. *et al.* Defining diurnal fluctuations in mouse choroid plexus and CSF at high molecular, spatial, and temporal resolution. *Nat Commun* **14**, doi:ARTN 3720
10.1038/s41467-023-39326-3 (2023).
- 17 Buxbaum, J., Koziol, J. & Connors, L. H. Serum transthyretin levels in senile systemic amyloidosis: effects of age, gender and ethnicity. *Amyloid* **15**, 255-261, doi:10.1080/13506120802525285 (2008).
- 18 Hanson, J. L. S. *et al.* Use of Serum Transthyretin as a Prognostic Indicator and Predictor of Outcome in Cardiac Amyloid Disease Associated With Wild-Type Transthyretin. *Circ-Heart Fail* **11**, doi:ARTN e004000
10.1161/CIRCHEARTFAILURE.117.004000 (2018).

REVIEWERS' COMMENTS

Reviewer #1 (Remarks to the Author):

The authors have done a diligent job in addressing all my previous comments. All my statistical comments/queries have been well addressed. I congratulate them on the thoroughness of their responses and revisions.

James Brophy MD PhD

Professor of Medicine & Epidemiology

McGill University

Reviewer #2 (Remarks to the Author):

I was asked to re review the revised version of the manuscript 'Transthyretin Levels, Clinical Correlates, and Adverse Clinical Outcomes [NCOMMS-24-17577A]'.

The present manuscript by Dr. Naman S. Shetty and Colleagues, offers interesting insights on the relationship between TTR levels and long-term outcome based on the UK Biobank registry.

The Authors have replied to the Statistical Reviewer observations and our comments. Overall, the updated results presented offer interesting insight into the potential role of TTR quantification, even more so with adjustment by gender (with women bearing higher TTR values) and age brackets.

With respect to the Results and the Discussion, within the editorial limits, the Authors could speculate more on gender and the potential pathophysiological link between levels of TTR and outcome.

Reviewer #3 (Remarks to the Author):

Response to Review

REVIEWERS' COMMENTS

Reviewer #1 (Remarks to the Author):

The authors have done a diligent job in addressing all my previous comments. All my statistical comments/queries have been well addressed. I congratulate them on the thoroughness of their responses and revisions.

**James Brophy MD PhD
Professor of Medicine & Epidemiology
McGill University**

Response: We greatly appreciate Dr. Brophy's thorough review of our manuscript. We believe that his comments and feedback have greatly improved the quality of our work.

Reviewer #2 (Remarks to the Author):

I was asked to re review the revised version of the manuscript ‘Transthyretin Levels, Clinical Correlates, and Adverse Clinical Outcomes [NCOMMS-24-17577A]’.

The present manuscript by Dr. Naman S. Shetty and Colleagues, offers interesting insights on the relationship between TTR levels and long-term outcome based on the UK Biobank registry.

The Authors have replied to the Statistical Reviewer observations and our comments. Overall, the updated results presented offer interesting insight into the potential role of TTR quantification, even more so with adjustment by gender (with women bearing higher TTR values) and age brackets.

Response: We appreciate the Reviewer’s suggestions. We have incorporated the additional feedback that we received from the Reviewer in our revised manuscript.

With respect to the Results and the Discussion, within the editorial limits, the Authors could speculate more on gender and the potential pathophysiological link between levels of TTR and outcome.

Response: We thank the Reviewer for their comment. As suggested by the Reviewer, we have further discussed the sex-based differences in transthyretin levels and the pathophysiological link between TTR levels and outcomes in our revised work.

Page 7 Line 119

While the precise mechanism of sex differences in TTR levels has not been ascertained, it could be postulated that sex hormones may play a role in altering TTR production. Sex hormones have been shown to regulate hepatic TTR expression in animal models, with testosterone inducing a larger increase in TTR synthesis than estrogen.¹

Page 8 Line 140

Although the underlying mechanism by which reduced TTR levels increase the risk of adverse outcomes is not known, it could be postulated that decreased TTR levels reflect TTR protein instability. Prior literature has demonstrated that individuals carrying the V142I TTR destabilizing genetic variant have an increased risk of heart failure due to deposition of misfolded TTR fibrils in the heart.²⁻⁵ Therefore, individuals with lower TTR levels could be hypothesized to have higher TTR instability, which in turn, increases their risk of adverse clinical outcomes.

Reviewer #3 (Remarks to the Author):

Response: We thank the Reviewer for their contributions in the review process.

References

- 1 Gonçaves, I. *et al.* Transthyretin is up-regulated by sex hormones in mice liver. *Mol Cell Biochem* **317**, 137-142, doi:10.1007/s11010-008-9841-2 (2008).
- 2 Parcha, V. *et al.* Association of Transthyretin Val122Ile Variant With Incident Heart Failure Among Black Individuals. *JAMA* **327**, 1368-1378, doi:10.1001/jama.2022.2896 (2022).
- 3 Coniglio, A. C. *et al.* Transthyretin V142I Genetic Variant and Cardiac Remodeling, Injury, and Heart Failure Risk in Black Adults. *JACC Heart Fail* **10**, 129-138, doi:10.1016/j.jchf.2021.09.006 (2022).
- 4 Quarta, C. C. *et al.* The amyloidogenic V122I transthyretin variant in elderly black Americans. *N Engl J Med* **372**, 21-29, doi:10.1056/NEJMoa1404852 (2015).
- 5 Damrauer, S. M. *et al.* Association of the V122I Hereditary Transthyretin Amyloidosis Genetic Variant With Heart Failure Among Individuals of African or Hispanic/Latino Ancestry. *JAMA* **322**, 2191-2202, doi:10.1001/jama.2019.17935 (2019).